# Poor adult tuberculosis treatment outcome and associated factors in Gibe Woreda, Southern Ethiopia: An institution-based cross-sectional study

**Melkamu Merid Mengesha**[1], **Mathewos Alemu Gebremichael**[1]*, **Desta Watumo**[2], **Inger Kristensson Hallström**[3], **Degu Jerene**[3,4]

**1** Department of Epidemiology and Biostatistics, School of Public Health, College of Medicine and Health Sciences, Arba Minch University, Arba Minch, Ethiopia, **2** Hadiya Zone Health Department, Hosanna, Ethiopia, **3** Faculty of Medicine, Department of Health Sciences, Child and Family Health, Lund University, Lund, Sweden, **4** KNCV Tuberculosis Foundation, Den Haag, The Netherlands

* alemumathewos2017@gmail.com

**Data Availability Statement:** All relevant data are available on the paper and the anonymized STATA

## Abstract

Tuberculosis (TB) remains a major medical and public health problem throughout the world, especially in developing countries including Ethiopia. Its control program is currently being challenged by the spread of drug-resistant TB, which is the result of poor treatment outcomes. Hence, this study assessed poor adult TB treatment outcomes and associated factors in Gibe Woreda, Southern Ethiopia. An institution-based cross-sectional study was conducted from March 1, 2020 to March 30, 2020, using a standard checklist to review clinical charts of TB patients who enrolled on first-line TB treatment under DOTS between June 2016 and June 2019. Poor treatment outcomes constituted death during treatment, treatment failure, and loss to follow-up (LTFU). Descriptive statistics were used to describe the characteristics of study participants. A binary logistic regression model was fitted to identify factors influencing treatment outcome and adjusted odds ratios with a 95% confidence interval were reported. The statistical significance of all tests in this study was declared at P-value <5%. A total of 400 adult TB patients were participated. The mean age of study participants was 39.2±16.7 years, 55.5% were males and 79.8% were pulmonary tuberculosis cases. Regarding the treatment outcomes, 58% completed treatment, 27.5% cured, 9.3% were LTFU, 3.2% died, and 2.0% failed. The overall poor treatment outcome was 14.5% (95% CI: 11.1–17.9). Age (aOR = 1.02; 95%CI: 1.01–1.04), male gender (aOR = 1.82; 95% CI: 0.99–3.73), travel $\geq$ 10 kilometres to receive TB treatment (aOR = 6.55; 95% CI: 3.02–14.19), and lack of family support during the course of treatment (aOR = 3.03; 95% CI: 1.37–6.70), and bedridden baseline functional status (aOR = 4.40; 95% CI: 0.96–20.06) were factors associated with poor treatment outcome. Successful TB treatment outcome in this study area was below the national TB treatment success rate. To improve positive treatment outcomes, remote areas should be prioritized for TB interventions, and stakeholders in TB treatment and care should give special emphasis to adults over the age of 45 years, males, those who travel more than 10 kilometres to receive TB care, having bedridden baseline functional status and those who had no family support.

data set of study participants was uploaded as a supporting information file.

**Funding:** WD received funding for research data collection from Haramaya University, College of Health and Medical Sciences. The funder had no role in study design, data collection and analysis, decision to publish, or preparation of the manuscript.

**Competing interests:** The authors declare that they have no competing interests.

**Abbreviations:** aOR, adjusted odds ratio; CI, confidence interval; DOTS, directly observed treatment short course; FMOH, Federal Ministry of health; HIV, human immune virus; IHRERC, institutional health research ethics committee; MDR-TB, Multidrug resistance Tuberculosis; MOH, Minster of health; MTB, Mycobacterium Tuberculosis; TB, Tuberculosis bacilli; TBL, Tuberculosis leprosy; UNICEF, united nation international children fund; UNAIDS, United Nations program on HIV/AIDS; WHO, world health organization.

# Background

Tuberculosis (TB) is a communicable disease caused by *Mycobacterium tuberculosis* that spreads when people sick with TB expel bacteria into the air [1]. According to 2017 World Health Organization (WHO) report, TB is a leading single infectious disease placed among the top 10 causes of death globally [1,2]. Pulmonary TB affects the lungs, and when TB affects other body sites, it is called extrapulmonary TB [1,2]. Cough, weakness, weight loss, fever, sweating, and swellings are among the commonly reported clinical manifestations of TB [1,2].

According to the 2020 World Health Organization (WHO) global TB report, about a quarter of the global population is infected with *Mycobacterium tuberculosis*, an estimated 10.0 million people fell ill with TB, and 1.2 million people died in 2019 [3]. Africa, as a continent, shared an estimated 25% of the global TB cases [3]. Ethiopia, a country in East Africa, ranked 10[th] among the 30 high TB-burden countries with an estimated TB incidence of 140 cases per 100,000 population [3]. Furthermore, the burden of multidrug-resistant tuberculosis (MDR-TB) in Ethiopia has been increasing from 1.6% to 2.3%, and the burden of MDR-TB among new and previously treated TB was 11.8% vs 17.8%, respectively [4]. Though 85% of TB can be successfully treated with a 6-months first-line regimen [3], the emergence of antimicrobial resistance has become a significant challenge to the gains achieved in the fight against TB [5]. Research evidence showed that previous exposure to or unsuccessful TB treatment was a major risk factor for the development of MDR-TB [6].

Performance of TB control programs is measured by five exclusive TB treatment outcome categories including cure rate, treatment completion, treatment failure, death during treatment, and LTFU [7,8]. From 2003 to 2016, the successful TB treatment outcomes (cure and treatment completion) varied between 81.1% and 86.3% in Ethiopia [9]. Subgroup analysis by regional States in Ethiopia revealed that successful treatment outcomes varied from 83.8%–94.2% in Afar, 82.6%– 94.5% in Oromia, 84.4%–87.9% in Gambella, 62.1%-97.9% in Tigray, and 74.5% -87.4% in Amhara [9]. Similarly, poor treatment outcomes (failure, death, and LTFU) at the national level varied between 7.5 to 26% in Ethiopia over the years, 2017 and 2020 [10].

Previous research indicated different factors associated with poor TB treatment outcomes. These included: socioeconomic and demographic variables (age, sex, residence, low income, limited access to transport, distance from home to the treatment centre, and limited social support) [9–19]. Clinical factors (type of tuberculosis, TB/HIV comorbidity, previous tuberculosis treatment, multidrug resistance), healthcare and environmental-related factors (year of registration, higher annual mean temperature per degree Celsius and health centre as a treatment access point), and individual factors (not knowing the HIV status, pre-treatment weight, and poor knowledge about TB) were factors previously identified to have associated with poor treatment outcomes [9–19].

Despite intensive efforts to control the risk factors for treatment success, TB remains a major public health issue in low-income countries including Ethiopia [10]. It is important to investigate context-specific TB treatment outcomes and the possible associated factors which will then aid to prevent the occurrence of further adverse outcomes. A national study that mapped poor TB treatment outcome in Ethiopia reported the regional rate for Southern Ethiopia, without identifying local values within the region, was 8.3–15.3% with a national average of 9.0% [12]. The authors believed that it was wise to identify pocket areas with significant poor TB treatment outcome in the region with poor TB treatment outcome of well above the national average [12]. Furthermore, a recent meta-analysis and systematic review of poor TB treatment outcome in Ethiopia identified only three studies that represented the regional estimation implying that more research was needed on the topic to have a good picture of poor

treatment outcome [9]. Hence, this study assessed the level of poor TB treatment outcomes and associated factors in Gibe Woreda, Southern Ethiopia.

## Methods

### Study area and period

The study was conducted in four public health facilities in Gibe Woreda (Amboro Health Centre, Homacho Primary Hospital, Megacho Health Centre, and Omochora Health Centre) in Hadiya zone, Southern Ethiopia. Homacho is the capital town of the Woreda located 262 Kilometres far from Addis Ababa the capital city of Ethiopia. The Woreda had a total population of 141,061. The public health facilities in the Woreda provide services in various outpatient and inpatient departments including follow-up and treatments for TB patients. Diagnosis and treatments of TB are based on the national TB treatment guideline of Ethiopia [20]. During the period from 2016 to 2019, a total of 1211 TB patients received TB care in the four health facilities mentioned above (Amboro Health Centre, 232; Homocho Primary Hospital, 318; Megacho Health Centre, 314; and Omochora Health Centre, 314). Three years retrospective data from June 2016 to June 2019 was collected from 1st March to 30th March 2020.

### Study design and population

Health facility-based cross-sectional study was conducted by reviewing clinical charts of TB patients who enrolled in first-line TB treatment under DOTS between June 2016 and June 2019. All adults ($\geq$15 years) who enrolled and received TB care in the four health facilities in Gibe Woreda with their TB treatment outcomes documented in the registration logbook were included. Adult patients who transferred out to another facility were excluded.

### Sample size determination and sampling procedure

This study used data collected for another project and used the same sample size determined for the original study (unpublished) focused on loss-to-follow-up from TB treatment. The sample size considered in the original study was 402. The assumptions considered for the sample size calculations were 5% significance level, 90% power, 1:1.6 allocation ratio (exposed group (patients travelled 10 km and above to receive TB treatment) to an unexposed group (patients travelled less than 10 km to receive TB treatment)), and a reference hazard ratio of 1.4 [21] for the association between distance travelled to the nearest health facility and loss-to-follow-up from TB treatment. In the original study, simple random sampling technique per the facilities studied was used to proportionally select TB patients from list of 1211 eligible adult TB patients in the four facilities who received first line TB-treatment between June 2016 and June 2019.

### Data collection procedure and study variables measurement

Data were collected by reviewing patient records using a standard checklist prepared from the variables available in the registration log book and patient cards. The data extraction tool included variables on socio-economic and demographic characteristics (age, sex, residence, marital status, occupational status, educational status, family support, and religion), treatment-related (type of TB, previous TB treatment, DOTS follow-up centre, treatment outcome, HIV/AIDS status, and nutrition support), and individual and healthcare-related (distance travelled and functional status). Two diploma nurses working in each of the clinics in the selected facilities collected the data and one of the investigators supervised the overall data collection process.

TB diagnosis and treatment were made according to the Ethiopian National Guidelines on TB, Drug-resistant TB, and Leprosy [22]. A smear-positive pulmonary TB (PTB+) was defined when a patient has positive acid-fast bacilli (AFB) results for at least one or two initial sputum specimens by direct microscopy. Diagnosis for smear-negative pulmonary TB (PTB⁻) was established when direct microscopy indicated two AFB negative results, and no response to a course of broad-spectrum antibiotics, and radiological abnormalities consistent with pulmonary TB, and decision by a clinician to treat with a full course of anti-TB or patient whose diagnosis is based on culture positive for *Mycobacterium tuberculosis*. When there is strong clinical evidence that TB has affected body organs other than the lungs, a patient is diagnosed with extra-pulmonary tuberculosis (EPTB) and a physician decides to treat the patient with a full course of anti-TB therapy. During the first two months, (intensive phase) patients receive daily rifampicin, pyrazinamide, isoniazid, and ethambutol followed by daily rifampicin and isoniazid for 4 months or more in the continuation phase.

Treatment outcomes included cure (confirmed smear-negative in the last month of treatment and on at least one previous occasion), treatment completed (a patient completed treatment and had no evidence of failure but without records to evidence cure), treatment failure (a patient whose sputum smear or culture is positive at month 5 or later during treatment), died (patient who dies during TB treatment), and LTFU (a patient who has been on TB treatment for at least four weeks and whose treatment was interrupted for eight or more consecutive weeks). Treatment success was finally defined as a sum of cured and completed treatment and poor treatment outcome was defined as a sum of failure, death, and loss to follow up [11,12].

Body mass index (BMI) was used to measure nutritional status, and categorized as severely underweight—BMI < 16.5kg/m², underweight—BMI under 18.5 kg/m², normal weight–BMI ≥ 18.5 to 24.9 kg/m², overweight – BMI ≥25 to 29.9 kg/m², and obesity – BMI ≥30 kg/m² [23]. In the present study, a major side effect was defined when there was any adverse reaction that resulted in discontinuation of the anti TB drugs, and/or directly resulted in hospitalization such as peripheral neuropathy, anaemia, hepatitis, hearing impaired, acute kidney injury [7]. Anti-TB treatment/medication adherence was measured, and dichotomized as adherent and non-adherent. A TB patient in either the intensive or continuation phase under a new or retreatment regimen, and who missed ≥ 10% of the total prescribed TB medication dose was considered as non-adherent [8].The functional status was measured according to the WHO patient monitoring guidelines, working: "able to perform usual work in or out of the house, harvest, go to school or, for children, normal activities or playing;" ambulatory: "able to perform activities of daily living but not able to work or play;" and bedridden: "not able to perform activities of daily living [14]."

## Data processing and statistical analysis

After checking the completeness of the checklist, data were entered into Epi data version 3.1. For further management and analysis, data were exported data to STATA software version 14.0. Descriptive analysis was employed to describe characteristics of TB patients using frequencies, proportions, and numerical summary measures. Both bivariable and multivariable binary logistic regression were applied to determine the factors associated with the poor outcome of TB treatment. The poor TB treatment outcome was coded"1" and "0" for the successful treatment outcome. Variables with a p-value < 0.25 in the bivariable analysis were entered into the multivariable analysis. A crude and adjusted odds ratio (OR) with a corresponding 95% confidence interval (CI) were computed and reported. Multicollinearity was checked using the variance inflation factor (VIF); in this study, the mean and maximum VIF of variables included the model were 2.2 and 5.2, respectively. Model fitness was assessed by Hosmer

and Lemeshow's goodness of fit test and it was a good fit (p-value = 0.2). Statistical significance was declared at p-value < 0.05.

### Ethical approval and consent to participate

The study was approved by the Institutional Health Research Ethics Review Committee (IHRERC) of the College of Health and Medical Sciences, Haramaya University. As the study was based on record reviews, the IHRERC approved consent from the respective health facility heads to access patient data. Besides the ethical approval letters, the College of Health and Medical Sciences wrote support letter requesting cooperation of the respective health facilities. To maintain confidentiality, all patient data were collected anonymously without personal identifiers. The study was also conducted following the declaration of Helsinki.

## Results

### Socio-demographic characteristics

A total of 400 TB patients were included in the analysis. Males accounted for 55.5% of the sample and the mean age was 39.2 years (± 16.7 SD) with the minimum and maximum ages of subjects in the study being 15 and 79, respectively. TB patients from the rural areas accounted for 83% of the study subjects and 34% had no formal education (Table 1).

### Clinical, treatment-related, and behavioural characteristics

New TB cases accounted for 96% of all cases with 79.8% being pulmonary TB cases. Regarding co-infection with HIV/AIDS, 3.5% were co-infected with HIV/AIDS. During the TB treatment period, 87.3% reported that they received family and 20% received nutritional support. On TB treatment initiation, 61.7% had a working functional status and 67.5% had normal nutritional status (Table 2).

### Prevalence of poor TB treatment outcome

Of the total TB patients, 14.5% (95% CI: 11.1–17.9) had poor treatment outcomes. Among patients experienced poor treatment outcome, 9.3% were LTFU, 3.2% died, and 2.0% failed. Successful treatment outcomes constituted 85.5% where 27.5% were cured and 58.0% completed treatment. Treatment outcomes by complete year of assessment were 11.2% (16/143) in 2017, 16.9% (30/177) in 2018 and 16.4% (12/73) in 2019. Among adult patients, 11.5% of those who travelled ≥ 10 kilometres to receive TB treatment had poor treatment outcome, whereas only 3% those travelled less than 10 kilometres experienced poor TB treatment outcomes. Among adult TB patients who had no family support, 5.8% experienced poor TB treatment outcomes.

### Factors associated with poor TB treatment outcome

Variables with p-values < 0.25 in the bivariable binary logistic regression analysis entered the multivariable model. The variables entered in the multivariable binary logistic regression included age, sex, occupation status, residence, distance travelled to reach the health facility, baseline functional status, family support, sputum smear result, and cigarette smoking. Among these variables, age, distance travelled, family support was significantly associated to poor TB treatment outcome, and also sex and baseline functional status was marginally associated (Table 3).

A 2% increase in poor treatment outcomes was observed for a corresponding one year increase in age (age, aOR = 1.02; 95%CI: **1.01–1.04**). Similarly, a stronger association with

**Table 1. Socio-demographic characteristics among adult patients registered at Gibe Woreda public health facilities, Hadiya zone, Southern Ethiopia from 1st January 2017 to 31st December 2019.**

| Characteristics | | TB treatment outcome | | Total, |
|---|---|---|---|---|
| | | Poor, n (%) | Good, n (%) | n (%). |
| Age (in years) | 15–24 years | 6 (1.3) | 96 (24) | 102 (25.3) |
| | 25–34 years | 8 (2) | 72 (18) | 81 (20) |
| | 35–44 years | 6 (1.5) | 68 (17.0) | 74 (18.5) |
| | 45–54 years | 16 (4) | 39 (9.8) | 55 (13.8) |
| | 55–64 years | 12 (3) | 35 (8.75) | 47 (11.8) |
| | 65–77 years | 10 (2.5) | 32 (8) | 42 (10.5) |
| Sex | Male | 40 (10) | 182 (45.5) | 222 (55.5) |
| | Female | 18 (4.5) | 160 (40) | 178 (44.5) |
| Residence | Urban | 19 (4.8) | 49 (12.2) | 68 (17) |
| | Rural | 39 (9.8) | 293 (73.2) | 332 (83) |
| Educational status | No formal education | 19 (4.8) | 117 (29.2) | 136 (34) |
| | Primary level | 16 (4) | 94 (23.5) | 110 (27.5) |
| | Secondary level | 16 (4) | 87 (21.8) | 103 (25.8) |
| | College and above | 7 (1.8) | 44 (11) | 51 (12.8) |
| Occupation status | Civil servant | 2 (0.5) | 33 (8.3) | 35 (8.8) |
| | Private | 39 (9.8) | 179 (44.8) | 218 (54.5) |
| | Unemployed[a] | 17 (4.3) | 130 (32.5) | 147 (36.8) |
| Religion | Protestants | 37 (9.3) | 248 (62) | 285 (71.3) |
| | Orthodox | 19 (4.8) | 87 (21.7) | 106 (26.5) |
| | Muslims | 2 (0.5) | 7 (1.8) | 9 (2.3) |
| Marital status | Never married | 19 (4.8) | 125 (31.2) | 144 (36) |
| | Married | 38 (9.5) | 210 (52.5) | 248 (62) |
| | Currently not married | 1 (0.3) | 7 (1.9) | 8 (2.2) |
| Weight (in kg) | < 35 kg | 6 (1.5) | 29 (7.3) | 35 (8.8) |
| | ≥ 35 kg | 52 (13) | 313 (78.2) | 365 (91.2) |
| Disclosure of TB status | Disclosed | 37 (9.3%) | 298 (74.5%) | 335 (83.8) |
| | Not disclosed | 21 (5.3%) | 44 (11%) | 65 (16.2) |
| Distance to health facility | Less than 10 km | 12 (3) | 239 (59.8) | 251 (62.8) |
| | 10 km and above | 46 (11.5) | 103 (25.8) | 149 (37.3) |

[a] = Housewives, and students, Kg: Kilograms, TB: Tuberculosis.

poor TB treatment outcome was observed among adults who travelled over 10 kilometres to reach the nearest health facility to receive TB care (travel ≥ 10 kilometres to receive care, aOR = 6.55; 95% CI: 3.02–14.19). Another strong association was observed among patients who lacked family support (Lack of family support, aOR = 3.03; 95% CI: **1.37–6.70)** (Table 3).

## Discussion

This study found an overall level poor TB treatment outcome of 14.5% (95% CI: 11.1–17.9) in Gibe Woreda, South Ethiopia. The factors identified to have significantly associated with poor TB treatment outcomes included age, sex, distance from the nearest health facility, lack of family support, and baseline functional status.

The level of poor TB treatment outcomes observed in this study was 14.5%. This was comparable to findings from similar studies in southern Ethiopia, 14.8% in Dilla Town [14], and

**Table 2. Clinical and behavioural characteristics among adult patients registered at Gibe Woreda public health facilities, Hadiya zone, Southern Ethiopia from 1st January 2017 to 31st December 2019.**

| Characteristics | | TB treatment | | Total, n (%). |
|---|---|---|---|---|
| | | Poor, n (%). | Good, n (%). | |
| Type of TB | Pulmonary | 52 (13) | 267 (66.8) | 319 (79.8) |
| | Extra pulmonary | 6 (1.5) | 75 (18.7) | 81 (20.2) |
| Smear result | Positive | 25 (6.3) | 108 (27) | 133 (33.3) |
| | Negative | 33 (8.2) | 234 (58.5) | 267 (66.7) |
| Patient category | New | 49 (13.8) | 335 (82.2) | 384 (96) |
| | Relapse | 3 (0.7) | 13 (3.3) | 16 (4) |
| HIV status | Positive | 3 (0.8) | 11 (2.7) | 14 (3.5) |
| | Negative | 55 (13.8) | 331 (82.7) | 370 (92.5) |
| Baseline | Working | 40 (10) | 207 (51.8) | 247 (61.7) |
| functional status[a] | Ambulatory | 14 (3.5) | 130 (32.5) | 144 (36) |
| Recent functional status | Bedridden | 4 (1) | 5 (1.3) | 9 (2.3) |
| | Working | 32 (8) | 198 (49.5) | 230 (57.5) |
| | Ambulatory | 26 (6.5) | 144 (36) | 170 (42.5) |
| Baseline nutritional status[b] | Severely underweight | 6 (1.5) | 31 (7.8) | 37 (9.3) |
| | Underweight | 16 (4) | 77 (19.2) | 93 (23.2) |
| | Normal | 36 (9) | 234 (58.5) | 270 (67.5) |
| Recent nutritional status | Severely underweight | 3 (0.8) | 1 (0.2) | 4 (1.0) |
| | Underweight | 12 (3) | 77 (19.3) | 89 (22.3) |
| | Normal | 43 (10.8) | 264 (66) | 307 (76.7) |
| TB patient contact history[c] | Yes | 7 (1.8) | 32 (8) | 39 (9.8) |
| | No | 51 (12.7) | 310 (77.5) | 361 (90.2) |
| Smoking cigarette (current) | Yes | 8 (2) | 24 (6) | 32 (8) |
| | No | 50 (12.5) | 318 (79.5) | 368 (92) |
| Drinking alcohol (current) | Yes | 6 (1.5) | 28 (7) | 34 (8.5) |
| | No | 52 (13) | 314 (78.5) | 366(91.5) |
| TB medication adherence | Adhered | 55 (13.8) | 333 (83.3) | 388 (97) |
| | Not adhered | 3 (0.7) | 9 (2.3) | 12 (3) |
| Major adverse side effect | Yes | 2 (0.5) | 6 (1.5) | 8 (2) |
| | No | 56 (14) | 336 (84) | 392 (98) |
| Family support | Yes | 35 (8.8) | 314 (78.5) | 349 (87.3) |
| | No | 23 (5.7) | 28 (7) | 51 (12.7) |
| Nutritional support | Yes | 22 (5.5) | 58 (14.5) | 80 (20) |
| | No | 36 (9) | 284 (71) | 320 (80) |

HIV: Human Immune deficiency virus, TB: Tuberculosis

[a] **working**: "able to perform usual work in or out of the house, harvest, go to school or, for children, normal activities or playing", **ambulatory**: "able to perform activities of daily living but not able to work or play" and **bedridden**: "not able to perform activities of daily living"

[b] severely underweight (BMI < 16.5kg/m$^2$), underweight (BMI < 18.5 kg/m$^2$), and normal weight (BMI $\geq$ 18.5 to 24.9 kg/m$^2$)

[c] a person who shared the same enclosed living space for one or more nights or for frequent or extended periods during the day with the index case during the 3 months before the diagnosis of TB

[d] patients who got social support from families in remembering to take their medication, food, and financial assistance.

17.5% in Wolayta Sodo [15]. A systematic review and meta-analysis on TB treatment outcomes in Ethiopia that included studies published between 2003 and 2016 reported a 16.3% poor TB treatment outcome at a national level [9]. But, the current finding was higher than others have reported previously in Ethiopia which ranged from 5.2–10.1% [11,13,18,19,24]. The finding in

**Table 3. Factors associated with poor treatment outcomes of tuberculosis among adult patients registered at Gibe Woreda public health facilities, Hadiya zone, Southern Ethiopia from 1st January 2017 to 31st December 2019.**

| Variables | | cOR (95% CI) | aOR (95%CI) |
|---|---|---|---|
| Age (in years) | | 1.04 (1.02–1.05) | **1.02 (1.01–1.04)**\* |
| Sex | Male | 2.0 (1.1–3.54) | 1.82 (0.99–3.73) |
| | Female | Ref. | Ref. |
| Occupation | Civil servant | 0.46 (0.10–2.11) | 0.49 (0.09–2.59) |
| Status | Private | 1.67 (0.90–3.08) | 1.16 (0.55–2.56) |
| | Unemployed | Ref. | Ref. |
| Residence | Urban | 2.9 (1.6–5.4) | 1.63 (0.78–3.50) |
| | Rural | Ref. | Ref. |
| Distance to health | Less than 10 km | Ref. | Ref. |
| facility | 10 km and above | 8.9 (4.5–17.5) | **6.55 (3.02–14.19)**\* |
| Disclosure | Disclosed | Ref. | Ref. |
| of TB status | Not disclosed | 3.8 (2.1–7.2) | 0.5 (0.2–1.5) |
| Family support | Yes | Ref. | Ref. |
| | No | 7.4 (3.8–14.2) | **3.03 (1.37–6.70)**\* |
| Smear result | Positive | 1.6 (0.9–2.9) | 1.47 (0.74–2.91) |
| | Negative | Ref. | Ref. |
| Current cigarette Smoking | Yes | 2.1 (0.9–5.0) | 1.81 (0.62–5.26) |
| | No | Ref. | Ref. |
| Baseline functional | Working | Ref. | Ref. |
| status | Ambulatory | 0.6 (0.3–1.1) | 0.92(0.42–2.03) |
| | Bedridden | 4.1 (1.1–16.1) | 4.40 (0.96–20.06) |

\*p-value < 0.05, aOR: Adjusted odds ratio cOR: Crude odds ratio CI: Confidence interval, Ref.: Reference category, TB: Tuberculosis.

this study on treatment success rate, when compared to on the national rate as reported in a systematic review and meta-analysis [9], was comparable to the national average of 83.7%. The possible explanations for the observed differences could be due to the length of study period and higher sample size, differences in DOTS performance, more subjects from urban setting, and younger average age [11,13,18,19,24]. Studies showed that MDR-TB has been increasing in Ethiopia, especially in previously treated TB [4], and it became a significant challenge to the gains achieved in the fight against TB [5]. A poor/unsuccessful/ TB treatment outcome was a major risk factor for the development of MDR-TB [6]. Hence, fighting against poor/unsuccessful/ TB treatment outcome can contribute to the prevention of emerging MDR-TB.

As reported in a national level pooled analysis, there was unstable trend in treatment outcomes reported from different settings in Ethiopia [9]. Likewise, in this study, we observed small variations in the rate of poor treatment outcome over the years considered, excluding non-complete years: 11.2% in 2017, 16.9% in 2018 and 16.4 in 2019. Similar findings were reported in a study done conducted in Ethiopia that reported a lower unfavourable TB treatment outcome during the initial years considered in the study, but started to increase thereafter, and fall again [11]. This might be due to non-consistent attention from the side of program implementation where heightened attention would be there at initial periods and then fall until a new initiative comes in and improvement in patients' awareness to TB care.

We found that patient age was significantly associated with poor TB treatment outcomes. As age increases by one unit (a year), poor TB treatment outcomes increase by 2%. This finding was supported by the several reports from different settings [13,24]. As opposed to the

finding in our study, there were also studies that reported age had either no effect of younger age was a risk for poor treatment outcome [10,18]. The possible explanation for an increase in poor treatment outcome associated with an increase in age could be due to that increase may be associated with increases in the chance of having concomitant diseases, poor adherence, and general psychological deterioration, and become unable to reach health facility as different studies suggested [25,26].

In the present study, sex was marginally associated with poor TB treatment outcomes where males were two times more likely to had poor TB treatment outcomes than females. This finding was in agreement with previous similar studies conducted elsewhere [1,11,15,27]. This might be due to a relatively higher exposure of males to cigarette smoking, traveling a long distance for economic reasons and alcohol consumption as compared to females. However, this finding was contrary to the report of a study done in the Northeast Ethiopia Dessie and Woldia town [10,18,28].

The distance from a health facility/treatment centre showed a statistically significant association with the poor TB treatment outcome. TB patients who travelled more than 10 kilometres to reach the nearest health facility were 6.55 times more likely to develop a poor TB treatment outcome compared to those who travelled less. This finding was consistent with the study done in Adama City, Ethiopia which reported that TB patients near the treatment centre had a good TB treatment success rate as compared to their counterparts [16]. Besides, a systematic review and meta-analysis on non-adherence to anti-TB drugs in Ethiopia revealed that feeling long distance to health institutions was found to be an important determinant to non-adherence, implying consequence to failure to tuberculosis treatment [8].

Lack of family support during TB treatment was another factor associated with the poor TB treatment outcome. Patients who did not get family support were 3.03 times more likely to have poor TB treatment outcomes when compared to those patients who got family support in the TB treatment process. This might be because of lack of help in the transport cost, collection of medication, reminding the appointment and supervision in taking medication and providing emotional and psychological support. Previous similar studies in Ethiopia reported that patients on anti-TB treatment who had no people to remind them take their medications were more likely to forget to take medication, and time schedule to visit TB clinic, and more likely to be non-adherent [29,30].

Baseline functional status was also marginally associated with poor TB treatment outcome. Those adult TB patients who were bedridden at baseline were 4.40 times a higher risk of poor TB treatment outcome compared to patients who had working baseline functional status. This finding was in agreement with studies conducted in Ethiopia: Gondar [31], Bahir Dar [32], and Southwest Ethiopia [33,34]. In our study, among patients who were bedridden at baseline, about 55% were above the age of 45 years old. Evidences support that as the age increases, the chance of having chronic disease, inability to reach health facility, and psychological deterioration become common which synergistically may contribute to a poor TB treatment outcome [25,26].

As a limitation, we collected data from secondary sources primarily collected for reporting purposes which limited our analysis to variables only available in patient records. Furthermore, we did consider the health-facility level cluster effect as patients receiving treatment from a similar centre could have commonalities in service delivery or TB care.

## Conclusions

The current study showed that successful TB treatment outcome was below the national success rate. Hence, remote areas such as Gibe Woreda should be prioritized for TB interventions.

And also special emphasis and strict follow-up are required for the tuberculosis patients with older age, male sex, visiting the TB treatment centres from long distances,> 10 kilometres and above, bedridden baseline functional status, and did not receive family support during the TB treatment process to reduce poor TB treatment outcomes.

## Supporting information

**S1 Data. A data set for the poor TB treatment outcome and associated factors in Gibe Woreda, Southern Ethiopia.**
(RAR)

## Acknowledgments

The authors acknowledge Gibe Woreda Public Health Facilities for providing the access to the data. We also would like to acknowledge data collectors and supervisors for accomplishing their tasks.

## Author Contributions

**Conceptualization:** Melkamu Merid Mengesha, Desta Watumo.

**Data curation:** Melkamu Merid Mengesha, Desta Watumo.

**Formal analysis:** Melkamu Merid Mengesha, Mathewos Alemu Gebremichael, Desta Watumo.

**Funding acquisition:** Desta Watumo.

**Investigation:** Melkamu Merid Mengesha.

**Methodology:** Melkamu Merid Mengesha, Mathewos Alemu Gebremichael, Desta Watumo, Degu Jerene.

**Software:** Melkamu Merid Mengesha, Mathewos Alemu Gebremichael, Desta Watumo.

**Supervision:** Melkamu Merid Mengesha.

**Visualization:** Melkamu Merid Mengesha, Mathewos Alemu Gebremichael.

**Writing – original draft:** Melkamu Merid Mengesha, Mathewos Alemu Gebremichael.

**Writing – review & editing:** Melkamu Merid Mengesha, Mathewos Alemu Gebremichael, Inger Kristensson Hallström, Degu Jerene.

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
