## [Decision Letter · Decision Letter 0]

8 Oct 2021

PGPH-D-21-00626

Poor adult tuberculosis treatment outcome and associated factors in Gibe Woreda, Southern Ethiopia: an institution-based cross-sectional study

Dear Dr. Gebremichael,

Thank you for submitting your manuscript to PLOS Global Public Health. After careful consideration, we feel that it has merit but does not fully meet PLOS Global Public Health’s publication criteria as it currently stands. Therefore, we invite you to submit a revised version of the manuscript that addresses the points raised during the review process.

We look forward to receiving your revised manuscript.

Kind regards,

Leonardo Martinez

Academic Editor

Journal Requirements:

1. If you are reporting a retrospective study of medical records or archived samples, please ensure that you have discussed in the ethics statement whether all data were fully anonymized before you accessed them and/or whether the IRB or ethics committee waived the requirement for informed consent. If patients provided informed written consent to have data from their medical records used in research, please include this information.

2. In the online submission form, you indicated that "Data are available upon request from the corresponding author."

3. Please amend your detailed Financial Disclosure statement. This is published with the article, therefore should be completed in full sentences and contain the exact wording you wish to be published.

i) State the initials, alongside each funding source, of each author to receive each grant.

ii). State what role the funders took in the study. If the funders had no role in your study, please state: “The funders had no role in study design, data collection and analysis, decision to publish, or preparation of the manuscript.”

4. In your Financial Disclosure, you indicated "Funding was obtained from Haramaya University, College of health and medical sciences.", but in the Funding Information you indicated that no funding was received. Please revise the Funding Information field to reflect funding received.

Additional Editor Comments (if provided):

We have evaluated the critiques of your manuscript entitled "Poor adult tuberculosis treatment outcome and associated factors in Gibe Woreda, Southern Ethiopia: an institution-based cross-sectional study". Our reviewers raised concerns of which I agree. Specifically the reviewers mentioned issues with the multivariable model as currently presented including issues with sample size within variables, number of variables included, and whether to use a multilevel model. Consultation with a biostatistician may be useful here. Definitions of key variables should be included -- either in the main manuscript or a supplementary document. The lack of clarity regarding specific variables was mentioned by Reviewer 1 and 3 and is needed for clarity and interpretation of results. Lastly, the grammar and writing needs substantial work to improve interpretation and was mentioned most strongly by Reviewer 3. This includes all sections of the manuscript.

Reviewers' comments:

Reviewer's Responses to Questions

**Comments to the Author**

1. Does this manuscript meet PLOS Global Public Health’s publication criteria? Is the manuscript technically sound, and do the data support the conclusions? The manuscript must describe methodologically and ethically rigorous research with conclusions that are appropriately drawn based on the data presented.

Reviewer #1: Partly

Reviewer #2: Yes

Reviewer #3: Partly

2. Has the statistical analysis been performed appropriately and rigorously?

Reviewer #1: No

Reviewer #2: Yes

Reviewer #3: No

3. Have the authors made all data underlying the findings in their manuscript fully available (please refer to the Data Availability Statement at the start of the manuscript PDF file)?

Reviewer #1: No

Reviewer #2: Yes

Reviewer #3: No

4. Is the manuscript presented in an intelligible fashion and written in standard English?

Reviewer #1: Yes

Reviewer #2: Yes

Reviewer #3: No

5. Review Comments to the Author

Reviewer #1: In this manuscript, the authors use data from clinical charts of adult TB patients attending four centres in Gibe Woreda, Ethiopia, to describe treatment outcomes and assess factors associated with poor outcomes. Amongst 400 patients included in the analysis, 58 (14.5%) had poor outcomes, and the multivariable analysis suggested that such outcomes were more common amongst men, the oldest age group, those travelling further to the health facilities, and those without family support.

The results seem plausible, but I am concerned that the data available may not be able to support the multivariable analysis.

This is a sub-analysis of data collected for another study, so the sample size is based on the requirements for that study (n = 400). It is not clear how well this available sample size can address the objectives of this analysis. There are very few patients with poor outcomes in some groups, leading to very wide confidence intervals, particularly in the multivariable analysis. I am not sure how far this can be mitigated, but one or more of the following might help:

- For some variables, it might help somewhat if a different group is taken as the baseline (for example, for the age variable, taking the >=45 years group as the baseline as this has the largest number of participants).

- It may also be useful to reduce the number of categories for some variables, but only if it is meaningful to do so. For example, there are small numbers in several of the occupation categories and perhaps some of the groups could be combined. However, this might produce heterogeneous groups and the authors might not think this is appropriate.

- The number of variables in the multivariable model could be reduced. Currently there are at least 10 variables included (some with multiple categories), which is a lot given that there are only 52 events (patients with poor outcomes). For example, is it possible to select a few variables of primary interest (based on prior work and / or plausibility) rather than selecting variables for inclusion in the model based on p values in the univariable analysis?

Also regarding the analysis, did the authors consider adjusting for clustering by site, e.g. using multilevel logistic regression?

The text lists 12 variables which were included in the multivariable model, but two of these (nutritional support and type of TB) do not appear in Table 3.

The sample size section refers to 402 patients being included in the original study, although 400 are included in this analysis. What were the reasons for excluding the other two patients from this analysis?

Some of the categories in Table 1 need to be defined. For example:

- Are reactive and non-reactive (HIV status) equivalent to positive and negative?

- What are the definitions of severe, moderate and normal nutritional status?

- What is meant by having a contact person?

- How were good and fair medication adherence defined?

- What was considered a side effect?

- Does disclosure status refer to HIV disclosure? The text refers to HIV disclosure but the tables include data on disclosure status for all participants, not just those with HIV. Please clarify and define.

It would be helpful for Tables 1 and 2 to stratify by good / poor outcome status.

Minor comments

Line 69: the word “defaulter” is used – this is no longer a preferred term, I suggest replacing with lost to follow-up.

Line 155: I think there is a typo – variable inflation factor should be variance inflation factor?

Lines 196-199: please give the numbers (numerator and denominator) with poor outcomes each year as well as the percentages.

Line 201-202: “Among adult TB patients who had no family support (28), 23 (5.75 %) experienced poor TB treatment outcomes.” Based on Table 2, I think this might need to be rephrased – are you saying that of 51 patients who had no family support, 23 experience poor TB treatment outcomes? I am not sure what the 28 and 5.75% refer to.

I think the columns for occupation status in Table 3 may be the wrong way around (total of 342 in the poor outcome column and 58 in the successful column).

Reviewer #2: This is one of the important article as DRTB present major abstacles in the control of TB and it is very important to find out factors responsible for poor outcomes of DSTB as succesful outcomes of TB treatment may help in the cutt off the chain to convert to drug resistant TB. Good work

Reviewer #3: The manuscript “Poor adult tuberculosis treatment outcome and associated factors in Gibe Woreda,

Southern Ethiopia: an institution-based cross-sectional study” reports pulmonary TB treatment outcomes in a a district of Ethiopia. The description of patients may be helpful for the TB program and for regional comparisons, but unfortunately a lack of clarity regarding methods and results, and an underdeveloped discussion are significant weaknesses.

Major comments

The sample size determination and sampling procedure section starting in line 108 is confusing, particularly considering lines 104-105. In line 105 it indicates that all adults with TB were considered, but in line 114 it states that a sampling technique was considered. What was ultimately done? Was this a convenience sample with a target of 402, or were all adult TB patients included (except for those who transferred out)? Since the Results report 400 patients included in the analysis, it seems that this did not include all eligible patients. On what basis were the 400 patients chosen? The details reported in lines 109-113 lack context, and what does “exposed to an unexposed group” mean? This entire section needs to be revised for clarity.

Results: Out of how many eligible TB patients were the 400 chosen? Are there any indications how these differ from those who were not chosen?

Results: Why was age categorized with the highest category as 45+ years? This group had the highest frequency, making one wonder what the age distribution is above 45 years. Why not use age as a continuous variable in the analyses (could assess for linearity, use restricted cubic splines, etc.)? If age is kept in categories, I would strongly consider adding age categories with older ages. This would be helpful to interpret the Age results in the regression models. Similarly, why were body weight and distance travelled categorized, and on what basis were the cutoffs used?

Table 2: Some of the variables require more details (in footnote, or text, or supplementary material).

- What defines the categories of functional status variables, malnutrition categories?

- What does “Having contact person” mean?

- Do the smoking and drinking alcohol categories mean current? Ever? How much tobacco or alcohol use defines smoking or drinking? Any? A certain quantity per time?

- How is medication adherence defined? Is it self-report, directly-observed therapy / medication log? Pill count? What threshold defines adherence? Is it all medications, some medications? Is it adherence to anti-TB drugs? Some of them? All of them?

- What constitutes a side effect?

- What defines family support?

- What defines nutritional support?

It seems unlikely that there was 0% unsuccessful treatment outcome in 2016. How were the patients assigned to treatment year? Year of treatment start? Year of recorded TB treatment outcome? How many patients had treatment outcomes in 2016? Could it be that since the study period was June 2016 that very few people included in the study had an outcome recorded by the end of 2016 due to the duration of TB treatment? Similarly, how many treatment outcomes were recorded in 2019? Would it make more sense to compare 12 month study periods rather than by calendar year since the study period was June to June?

Analyses: With 58 poor TB treatment outcomes, it appears the adjusted model is at risk for being overfit based on the number of factors (and degrees of freedom) in the adjusted regression model.

Discussion: The Discussion is generally underdeveloped. It would be helpful to have a more robust discussion about how the findings of the study can support the local, regional, and national TB program efforts to improve TB treatment outcomes. To that end, reworking the very coarse categorization of Age and distance to health facility as continuous variables may help provide more targeted results for discussion. The Introduction also framed poor treatment outcomes in the context of drug-resistant TB (also noted in the Abstract). However, there is no mention of drug resistance in the Results or in the Discussion.

Minor comments

Lines 50-51: Would be helpful to update this and include the year so the reader knows when the statistic is relevant. The most recent WHO list of top 10 causes of deaths globally from 2019 no longer includes TB: https://www.who.int/news-room/fact-sheets/detail/the-top-10-causes-of-death, though the causes of death in the top 10 vary by country income status according to the WHO.

Line 69: Make sure the treatment outcome category names are used consistently. For example, in line 69 the outcome “defaulter” is listed, but in line 73, “lost to follow-up” is used. If referring to WHO outcomes, the term lost to follow-up is preferred, as noted in the document “Definitions and reporting framework for tuberculosis – 2013 revision (updated December 2014 and January 2020)”.

Line 69: Where did the successful TB treatment outcomes vary? Was this all of Ethiopia? Please clarify.

Lines 70-75: How do these subgroups relate to the Gibe Woreda noted in the title and which is the site of the study?

Lines 77-83: this is a long hierarchical list and it is confusing with multiple parentheses that do not all have a closing parenthesis. Please clarify and consider simplifying the text – perhaps with a table.

Line 88: Please clarify this sentence – what is the comparator for “poorer”? This is the first time in the main text that the authors have written “Gibe Woreda”. Perhaps in the paragraph with lines 69-75 the long list of values could be replaced with context-specific information about Gibe Woreda and how it fits in to Ethiopia and TB program outcomes. Then in the sentence in line 88, clearly identify what you are comparing Gibe Woreda to when you say that treatment “outcome is” (outcomes are?) poorer there.

Line 194: what does “about” mean? Is there uncertainty about this number?

Line 196: what does “indifferent enrolment periods” mean?

Table 3: what does Disclosure status mean? Earlier in the text and previous table are TB disclosure status and HIV disclosure status, so need to clarify.

Line 232: where are these “other similar settings”?

Lines 259-261: This sentence is confusing, please clarify.

Line 120: ART status mentioned, but not HIV status. Then ART status is not mentioned anywhere else.

Line 128: smear-negative

Line 157: if the statement “it was a good fit…” is a result of the goodness of fit tests, this should be recorded in the Results, not the Methods.

I would recommend reviewing the manuscript to correct errors in grammar. I would also encourage consistency throughout the manuscript. For example:

- In line 56 you should use lowercase ‘t’ for “Mycobacterium tuberculosis” as in line 49.

- Line 104, 274 uses lowercase ‘w’ for “woreda” but it is capitalized elsewhere.

- Loss to follow-up is written with and without dashes and abbreviation (LTFU) throughout.

- Confidence intervals are separated by a comma in some places and dashes in others.

6. PLOS authors have the option to publish the peer review history of their article (what does this mean?). If published, this will include your full peer review and any attached files.

**Do you want your identity to be public for this peer review?** For information about this choice, including consent withdrawal, please see our Privacy Policy.

Reviewer #1: No

Reviewer #2: **Yes: **Mazhar Ali Khan

Reviewer #3: No

---

## [Editor Report · Decision Letter 1]

22 Dec 2021

Poor adult tuberculosis treatment outcome and associated factors in Gibe Woreda, Southern Ethiopia: an institution-based cross-sectional study

PGPH-D-21-00626R1

Dear Dr. Gebremichael,

We're pleased to inform you that your manuscript has been judged scientifically suitable for publication and will be formally accepted for publication once it meets all outstanding technical requirements.

Within one week, you'll receive an e-mail detailing the required amendments. When these have been addressed, you'll receive a formal acceptance letter and your manuscript will be scheduled for publication.

An invoice for payment will follow shortly after the formal acceptance. To ensure an efficient process, please log into Editorial Manager at https://www.editorialmanager.com/pgph/ click the 'Update My Information' link at the top of the page, and double check that your user information is up-to-date. If you have any billing related questions, please contact our Author Billing department directly at authorbilling@plos.org.

Kind regards,

Leonardo Martinez

Academic Editor